# VL-SAM-V2: Open-World Object Detection with General and Specific Query Fusion

**Zhiwei Lin    Yongtao Wang**[*]
Wangxuan Institute of Computer Technology, Peking University, China
{zwlin, wyt}@pku.edu.cn

## Abstract

Current perception models have achieved remarkable success by leveraging large-scale labeled datasets, but still face challenges in open-world environments with novel objects. To address this limitation, researchers introduce open-set perception models to detect or segment arbitrary test-time user-input categories. However, open-set models rely on human involvement to provide predefined object categories as input during inference. More recently, researchers have framed a more realistic and challenging task known as open-ended perception that aims to discover unseen objects without requiring any category-level input from humans at inference time. Nevertheless, open-ended models suffer from low performance compared to open-set models. In this paper, we present VL-SAM-V2, an open-world object detection framework that is capable of discovering unseen objects while achieving favorable performance. To achieve this, we combine queries from open-set and open-ended models and propose a general and specific query fusion module to allow different queries to interact. By adjusting queries from open-set models, we enable VL-SAM-V2 to be evaluated in the open-set or open-ended mode. In addition, to learn more diverse queries, we introduce ranked learnable queries to match queries with proposals from open-ended models by sorting. Moreover, we design a denoising point training strategy to facilitate the training process. Experimental results on LVIS show that our method surpasses the previous open-set and open-ended methods, especially on rare objects.

## 1   Introduction

Deep learning has made significant breakthroughs in computer vision tasks, enabling substantial progress in object detection tasks. Traditional object detection approaches [34, 16] typically employ a closed-set paradigm, in which detection models are restricted to recognizing and locating objects that are seen in the training set. However, closed-set approaches have notable limitations, particularly in real-world scenarios where novel or unseen objects appear. For instance, in an autonomous driving system, if its object detector trained to identify vehicles and pedestrians encounters a novel object [22], such as a drone or a wild animal, it may either misclassify the object or fail to detect it, which can even lead to serious traffic accidents.

To address this challenge, open-world methods [12, 9, 14] have been proposed, allowing models to be more flexible by enabling them to handle novel and unseen objects. Open-world methods generally can be divided into two categories: open-set (or open-vocabulary) [14, 29, 11] and open-ended [25, 27]. Open-set methods can identify objects within a predefined object category list, which is provided by humans during the inference phase, even if unseen objects are present in the list. However, these open-set models cannot predict objects outside the predefined object category list, which limits their ability to discover new object categories automatically. For instance, if the

---

[*]Corresponding author.

39th Conference on Neural Information Processing Systems (NeurIPS 2025).

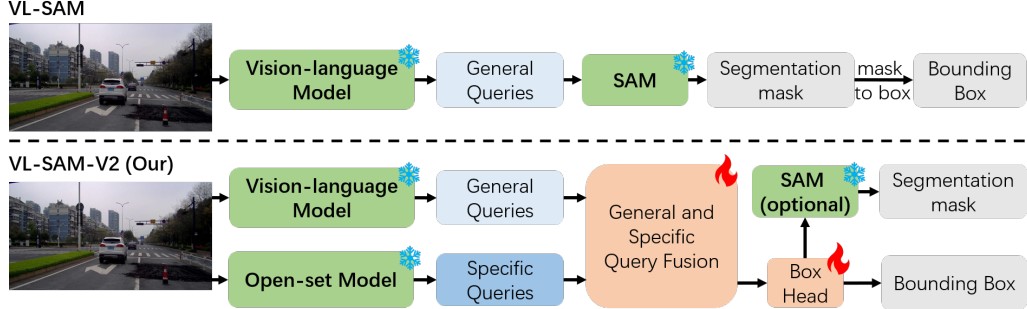

Figure 1: **Illustration of VL-SAM-V2.** VL-SAM-V2 combines the general queries from VL-SAM and the specific queries of an open-set model with a query fusion module.

predefined object category list only contains vehicles and pedestrians, the open-set models still cannot detect drones or wild animals in the input images. Therefore, they need humans to intervene by providing a more comprehensive list including these objects. Additionally, although open-set models perform well on frequent objects, they often struggle with rare objects [29]. In contrast, open-ended models aim to identify and categorize all objects in given images without requiring a predefined object category list provided by humans [25]. This capability is usually achieved by incorporating large vision-language models. However, open-ended models suffer from low performance [43, 27], especially when dealing with crowded objects, which typically appear in frequent object categories.

To this end, we introduce VL-SAM-V2, an open-world object detection framework that combines the strengths of open-set and open-ended methods to achieve favorable performance in both rare and frequent object categories while equipped with the ability to discover novel objects, as shown in Figure 1. Specifically, we treat object queries from open-set models as specific queries, since they specialize in frequent objects. Meanwhile, we adopt the VL-SAM pipeline to generate object queries in an open-ended manner and view them as general queries, because of the generalization of large vision-language models. Then, we propose a general and specific query fusion module to fuse two distinct types of queries with the attention mechanism in a lightweight manner. Since the VL-SAM only provides point prompts, we present ranked learnable queries by sorting the general point prompts based on their scores and matching them with the learnable queries to compose general queries. In addition, to facilitate the training process, we introduce the denoising points strategy by adding both positive and negative points as the noisy general queries. This denoising process helps models to learn to distinguish positive and high-accuracy general queries and to filter out negative queries or point prompts from VL-SAM. During inference, VL-SAM-V2 can be dynamically adapted as an open-set or open-ended model based on the predefined object category list. When the list is empty, VL-SAM-V2 is viewed as an open-ended model. Experimental results show that VL-SAM-V2 significantly improves detection performance on rare objects compared to existing open-set models by discovering novel objects with general queries. Meanwhile, VL-SAM-V2 compensates for the poor performance of open-ended models.

The main contributions of this work are summarized as follows:

- We introduce an open-world object detection framework, VL-SAM-V2, to combine the strengths of open-set and open-ended methods with the general and specific query fusion module. VL-SAM-V2 can be dynamically adapted as an open-set or open-ended model.
- We design ranked learnable queries to transform point prompts from VL-SAM into object queries and present a denoising points strategy to facilitate the training process.
- Experimental results show that VL-SAM-V2 achieves both state-of-the-art open-set and open-ended object detection performance on the LVIS dataset in a zero-shot manner.

## 2 Related work

### 2.1 Open-set Object Detection

Open-set or open-vocabulary object detection methods aim to detect objects in a predefined object category list given by humans [29]. To achieve this, current open-set object detection methods

often transform the open-set detection task into a vision-language alignment task, allowing detectors to match unseen object categories with visual information through the alignment process. With the advent of contrastive learning [15, 3], vision-language alignment models (*e.g.*, CLIP [33]) achieve favorable zero-shot alignment performance for unseen objects. Based on this, many open-set object detection methods use a proposal network to obtain foreground object bounding boxes and embeddings, and then use CLIP as the open-set classification module to predict their categories. However, CLIP is trained to align the whole image and texts, while the open-set object detection task favors region-level feature alignment.

To address this, GLIP [24] proposes an object-aware vision-language region-level alignment task with phrase grounding to pre-train open-world object detectors. DetCLIP [40] separates the texts in GLIP into a parallel concept and regards each object category as an individual input. GLIPv2 [44] further improves on GLIP by introducing a deep fusion block for better vision-language fusion and intra- and inter-image-text contrastive losses. GroundingDINO [29] follows the idea of GLIP to present image-text cross-modality fusion, and proposes object queries by text information. OWL [32] scales the training dataset to 10M examples with pseudo-annotation and a self-training strategy. YOLO-World [7] adopts the efficient YOLO-series backbone to achieve real-time open-set object detection. More recently, due to the emergence of large language models and vision-language models, many open-set methods try to adopt them for additional supervision. DetCLIPv3 [41] combines the object-level and image-level generation tasks with the open-set object detection task. LLMDet [11] explores two-stage co-training by generating image-level and region-level detail captions with a pre-trained vision-language model. In addition, some works [19, 43] directly fine-tune a pre-trained vision-language model to address the open-set object detection by following pix2seq [4].

However, the above methods require a predefined object category list given by humans and cannot discover novel objects by themselves. In this paper, we propose prompting the open-set method using point priority from large vision-language models and enabling the proposed model to discover novel objects not in the predefined object category list.

## 2.2 Open-ended Object Detection

Open-ended object detection methods can directly predict all objects in given images without any predefined object category list from humans. GenerateU [25] first introduces the open-ended problem and proposes a generative framework with a large language model to generate object categories from object queries. In addition, it constructs a large dataset with pseudo bounding boxes and caption pairs to fine-tune the whole model. To alleviate the training costs, VL-SAM [27] presents a training-free open-ended object detection and segmentation framework that connects a large vision-language model and the segment-anything model (SAM) [18] with attention maps as the intermediate prompts. Moreover, some large vision-language models [19, 43, 20] introduce specific tokens, like '<Det>', to allow models to predict objects in an open-ended manner. Though good grounding performance of these vision-language models is achieved, the accuracy of detecting all objects is very low.

Though some progress has been made on the open-ended detection task, the performance of current open-ended methods is significantly lower than that of open-set methods. To address this issue, we propose a general and specific fusion pipeline by fusing the queries from open-set and open-ended models. The proposed pipeline preserves the strengths of open-set methods' high performance and can discover novel objects like open-ended methods.

## 3 Method

### 3.1 Preliminary of VL-SAM

VL-SAM [27] is a training-free open-ended object detection and segmentation framework that connects vision-language and segment-anything models with attention maps as the intermediate prompts. It utilizes a vision-language model to generate the categories of all objects in the given image and stores the corresponding attention maps. Then, VL-SAM adopts the head aggregation and attention flow mechanism to aggregate and propagate attention maps through all heads and layers. After that, a sampling strategy is applied to the refined attention maps to generate positive and negative points as the point prompts for SAM. In addition, VL-SAM uses several ensemble strategies to obtain more comprehensive object points.

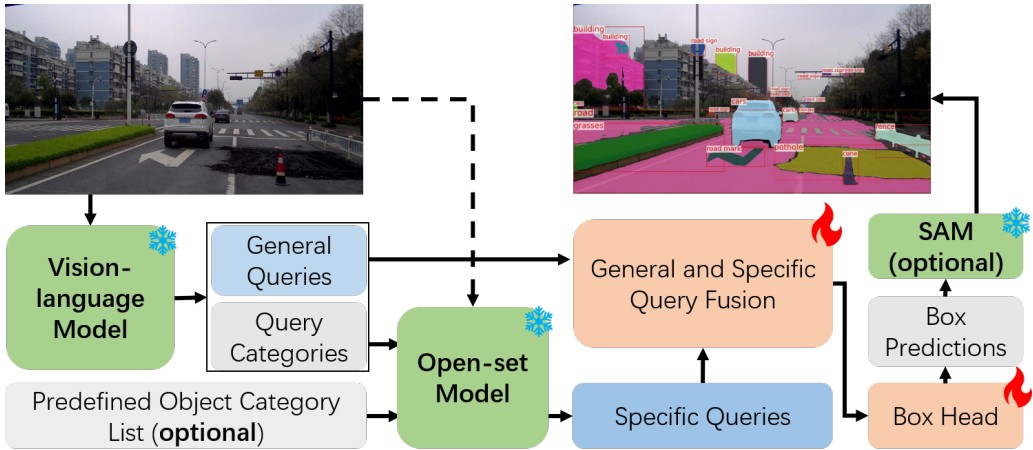

Figure 2: **The overall pipeline of VL-SAM-V2.** VL-SAM-V2 utilizes a vision-language model to generate general queries and a standard open-set detection model to generate specific queries. Then, the two distinct queries are sent to the general and specific query fusion module for interaction. Finally, a box head and an optional SAM are applied to predict the perception results. During the training, we only fine-tune the general and specific query fusion module and the box head. In addition, by controlling the predefined object category list, VL-SAM-V2 can operate in open-ended mode.

In this paper, we adopt VL-SAM to generate point prompts as proposals for the general queries in an open-ended manner.

### 3.2 VL-SAM-V2

As shown in Figure 2, we provide an overview of the proposed framework. Specifically, given an image input, we first use VL-SAM to discover all objects in the image and sample their point coordinates according to the attention maps. Then, we combine the discovered objects with the predefined object set to form the final object set for the open-set model to generate specific queries. Meanwhile, the sampled points are converted into initial bounding boxes and matched with learnable queries to compose general queries. After that, the general and specific queries are sent to the general and specific query fusion module to update the queries. Finally, the box head and optional SAM are applied to predict the final results.

**Denoising Points.** VL-SAM-V2 employs VL-SAM to generate point prompts to compose general queries. However, during training, the inference of VL-SAM will incur additional training costs due to the use of a large vision-language model. In addition, directly training with a few points from VL-SAM can lead to unstable training and poor convergence. Inspired by the denoising anchor boxes trick in the training of DETR [21, 42], we propose denoising points to address this issue. Specifically, we randomly sample positive points in the ground-truth bounding boxes and negative points outside the boxes to replace the point proposals generated by VL-SAM. The sampled noisy points are sent to the decoder, and the models are trained to denoise the noisy points into corresponding bounding boxes and labels. More concretely, we follow DINO [42] to sample points by adding coordinate noise $\Delta X$ and $\Delta Y$ with two hyper-parameters $\lambda_1$ and $\lambda_2$:

$$|\Delta X_p| < \lambda_1 \times w/2 < |\Delta X_n| < \lambda_2 \times w/2,$$
$$|\Delta Y_p| < \lambda_1 \times h/2 < |\Delta Y_n| < \lambda_2 \times h/2, \tag{1}$$

where $p$ and $n$ denote positive and negative points, $w$ and $h$ indicate the width and height of the ground-truth bounding box for denoising. The $\Delta X$ and $\Delta Y$ are sampled from the uniform distribution according to the range in Eq. 1.

To convert sampled noisy points into bounding boxes and general queries for subsequent fusion and decoding, we utilize multi-level image features to predict the initial bounding boxes according to the coordinates of noisy points and introduce ranked learnable queries. Specifically, given the coordinates of a noisy point $(X_{noise}, Y_{noise})$ and the multi-level image features $f$, we interpolate the image

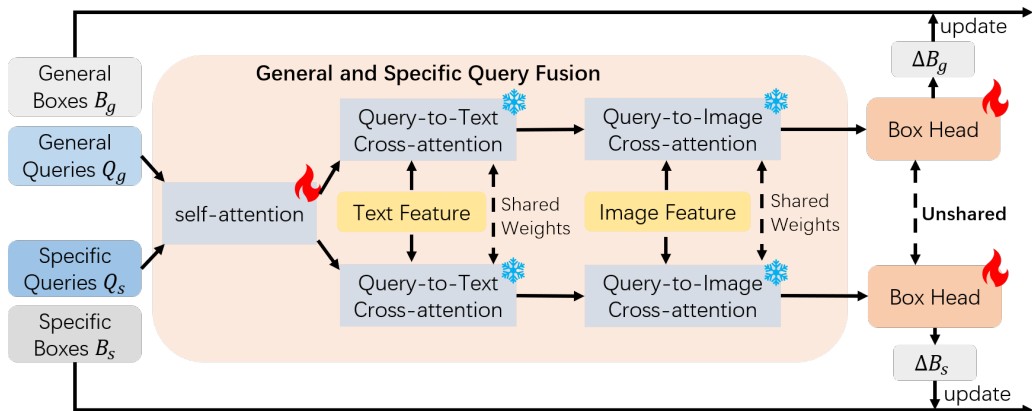

Figure 3: **Illustration of general and specific query fusion module.** General and specific queries interact with a self-attention mechanism. Then, the shared query-to-text and query-to-image cross-attention are applied for the two queries independently. Finally, the unshared box heads predict the offset of corresponding bounding boxes. During the training, we only update the parameters in the self-attention and box heads.

features in $(X_{noise}, Y_{noise})$ for all levels and obtain sampled point feature $f^l_{X_{noise}, Y_{noise}}$, where $l$ denotes the $l$-th level of the image features. After that, we send the point feature $f^l_{X_{noise}, Y_{noise}}$ to a linear layer and a layer normalization, followed by a box head in the open-set model to obtain the initial bounding box. The initial bounding box for each point is combined with the ranked learnable query to compose general queries and sent to the fusion module and box head.

**Ranked Learnable Queries.** Each point prompt or noisy point should be matched with a learnable query for subsequent fusion and decoding. However, point prompts or noisy points are sampled randomly without any order. A simple way is to set the same learnable query for each point, without considering their difference in coordinates or features. Intuitively, different queries are expected to handle different objects. Therefore, we introduce ranked learnable queries by matching different points with different queries. Specifically, for noisy points, we send $f^l_{X_{noise}, Y_{noise}}$ to the classification head to obtain the score of each noisy point. Then, we rank the noisy points from largest to smallest based on their scores. Meanwhile, we set $N$ additional learnable queries and fix their order after initialization. Finally, the ranked noisy points are matched with the learnable queries one by one. For example, the noisy point with the largest score matches the first learnable query. Since the number of noisy points may not match $N$, we simply discard the redundant noisy points or ranked learnable queries.

Similarly, during inference, the points sampled by VL-SAM can be ranked by scores and matched with ranked learnable queries.

**General and Specific Query Fusion.** As shown in Figure 3, after obtaining the initial bounding box and ranked learnable query of each point, we combine them with the bounding boxes and queries generated by the open-set model and send them into the general and specific query fusion module, which is distributed in each transformer decoder layer of the open-set model. The boxes and queries of sampled points are treated as general boxes $B_g$ and queries $Q_g$ since they are prompted by the large vision-language model, while the boxes and queries from the open-set model are denoted as specific boxes $B_s$ and queries $Q_s$. Since the number of sampled points varies, we adopt the self-attention module for the interaction between the general and specific queries, as shown in Figure 3. Specifically, $Q_g$ and $Q_s$ are first concatenated and sent to a standard self-attention module to obtain the fused queries. Then, the fused queries are separated and sent to a shared query-to-text cross-attention, a shared query-to-image cross-attention, and a shared FFN to obtain updated queries $\bar{Q}_g$ and $\bar{Q}_s$. After that, two unshared box heads are applied to $\bar{Q}_g$ and $\bar{Q}_s$ to predict the coordinate offsets of bounding boxes $\Delta B_g$ and $\Delta B_s$, respectively. Finally, we update the $B_g$ and $B_s$ as follows:

$$B_g \leftarrow B_g + \Delta B_g,$$
$$B_s \leftarrow B_s + \Delta B_s. \tag{2}$$

During training, we only update the parameters in the self-attention and box heads.

**Loss Function.** We adopt the grounding losses $\mathcal{L}_{grounding}$ and the generation losses $\mathcal{L}_{generation}$ in LLMDet [11] as the supervision to fine-tune VL-SAM-V2. The grounding losses contain vision-language alignment and bounding box regression losses, while the generation losses comprise region-level and image-level caption losses. In VL-SAM-V2, the alignment process between ground-truth and general/specific boxes is performed independently. The final loss can be calculated as follows:

$$\mathcal{L} = \frac{1}{2}(\mathcal{L}_{grounding}(\hat{B}, B_g) + \mathcal{L}_{grounding}(\hat{B}, B_s)) + \mathcal{L}_{generation}(\hat{T}, T), \tag{3}$$

where $\hat{B}$ denotes the ground-truth bounding boxes, $\hat{T}$ and $T$ are the ground-truth captions and generated text predictions from a vision-language model in LLMDet [11], respectively.

## 4 Experiments

### 4.1 Implementation Details

For VL-SAM, we choose InternVL-2.5-8B with InternViT-300M [6] and InternLM2.5-7B [5] as the vision-language model. We set the temperature to 0.8 and top-p for nucleus sampling to 0.8 for InternVL-2.5-8B. For the open-set model, we select LLMDet [11] as the baseline model because of its SOTA performance. The number $N$ of additional learnable queries is set to 900. For denoising points, both hyper-parameters $\lambda_1$ and $\lambda_2$ are set to 1.

The whole model of VL-SAM-V2 is fine-tuned with GroundingCap-1M dataset [11] following the training protocol of LLMDet [11]. During training, only the self-attention modules in general and specific query fusion and box heads are fine-tuned, while others are frozen. We fine-tune VL-SAM-V2 for 150k iterations using automatic mixed-precision with a batch size of 16. All training can be done on 8 NVIDIA A800 GPUs within two days.

### 4.2 Main Results

We mainly evaluate the proposed method on the LVIS dataset, which contains 1203 categories. We adopt the fixed AP [8] as the evaluation metric on frequent, common, and rare classes.

**Open-set Object Detection.** As shown in Table 1, VL-SAM-V2 beats all previous open-set models and achieves the new state-of-the-art zero-shot open-set object detection results. Specifically, with general and specific query fusion, VL-SAM-V2 outperforms the baseline LLMDet by 1.0 AP and 0.8 AP on LVIS minival with Swin-T [30] and Swin-L as backbones, respectively. Notably, for rare objects, VL-SAM-V2 obtains a significant performance improvement, from 37.3 $AP_r$ to 41.2 $AP_r$ with Swin-T and from 45.1 $AP_r$ to 47.2 $AP_r$ with Swin-L. These results demonstrate our motivation that the prior knowledge from large vision-language models enables the open-set model to discover rare objects.

In addition, we notice that DetCLIPv3 [41] achieves higher $AP_r$ and $AP_c$. The reason is that DetCLIPv3 collects more balanced data and noun concept corpora. Nevertheless, we believe that the performance of DetCLIPv3 can be further improved by the fusion pipeline of VL-SAM-V2.

Moreover, we evaluate VL-SAM-V2 on other datasets, including COCO [26] and CODA [22]. As shown in Table 3, our method achieves the best results on various datasets, especially on the CODA dataset.

**Open-ended Object Detection.** For open-ended object detection evaluation, we set the predefined object category list to empty and follow GenerateU [25] and VL-SAM [27] to match the generated object categories with the category list in LVIS by CLIP [33]. We present the open-ended object detection results in Table 2. The results show that VL-SAM-V2 achieves the best open-ended object detection performance on AP and $AP_r$. Specifically, with Swin-T as the backbone, VL-SAM-V2 outperforms GenerateU by 2.7 AP and 9.8 $AP_r$. For Swin-L, VL-SAM-V2 also improves $AP_r$ by a large margin compared to previous methods, from 22.3 $AP_r$ and 23.4 $AP_r$ to 30.5 $AP_r$. Moreover, though VL-SAM obtains a higher $AP_r$ than GenerateU, the overall AP is lower than GenerateU. This indicates that the priority of vision-language models favors rare objects but performs unsatisfactorily

Table 1: **Comparison of zero-shot open-set object detection results on LVIS val and minival [13].** We report *fixed* AP [8]. Grey results denote using additional private data.

| Method | Backbone | LVIS$^{minival}$ | | | | LVIS | | | |
|---|---|---|---|---|---|---|---|---|---|
| | | AP | AP$_r$ | AP$_c$ | AP$_f$ | AP | AP$_r$ | AP$_c$ | AP$_f$ |
| GLIP [24] | Swin-T | 26.0 | 20.8 | 21.4 | 31.0 | 17.2 | 10.1 | 12.5 | 25.2 |
| GLIPv2 [44] | Swin-T | 29.0 | – | – | – | – | – | – | – |
| CapDet [31] | Swin-T | 33.8 | 29.6 | 32.8 | 35.5 | – | – | – | – |
| Grounding-DINO [29] | Swin-T | 27.4 | 18.1 | 23.3 | 32.7 | 20.1 | 10.1 | 15.3 | 29.9 |
| OWL-ST [32] | CLIP B/16 | 34.4 | 38.3 | – | – | 28.6 | 30.3 | – | – |
| Desco-GLIP [23] | Swin-T | 34.6 | 30.8 | 30.5 | 39.0 | 26.2 | 19.6 | 22.0 | 33.6 |
| DetCLIP [40] | Swin-T | 35.9 | 33.2 | 35.7 | 36.4 | 28.4 | 25.0 | 27.0 | 28.4 |
| DetCLIPv2 [39] | Swin-T | 40.4 | 36.0 | 41.7 | 40.4 | 32.8 | 31.0 | 31.7 | 34.8 |
| YOLO-World-L [7] | YOLOv8-L | 35.4 | 27.6 | 34.1 | 38.0 | – | – | – | – |
| T-Rex2 [17] | Swin-T | 42.8 | 37.4 | 39.7 | 46.5 | 34.8 | 29.0 | 31.5 | 41.2 |
| OV-DINO [37] | Swin-T | 40.1 | 34.5 | 39.5 | 41.5 | 32.9 | 29.1 | 30.4 | 37.4 |
| LLMDet [11] | Swin-T | 44.7 | 37.3 | 39.5 | 50.7 | 34.9 | 26.0 | 30.1 | 44.3 |
| VL-SAM-V2 (Our) | Swin-T | **45.7** | 41.2 | 41.1 | 50.7 | **35.5** | 29.3 | 31.8 | 44.3 |
| GLIP [24] | Swin-L | 37.3 | 28.2 | 34.3 | 41.5 | 26.9 | 17.1 | 23.3 | 36.4 |
| GLIPv2 [44] | Swin-H | 50.1 | – | – | – | – | – | – | – |
| Grounding-DINO [29] | Swin-L | 33.9 | 22.2 | 30.7 | 38.8 | – | – | – | – |
| OWL-ST [32] | CLIP L/14 | 40.9 | 41.5 | – | – | 35.2 | 36.2 | – | – |
| DetCLIP [40] | Swin-L | 38.6 | 36.0 | 38.3 | 39.3 | 28.4 | 25.0 | 27.0 | 31.6 |
| DetCLIPv2 [39] | Swin-L | 44.7 | 43.1 | 46.3 | 43.7 | 36.6 | 33.3 | 36.2 | 38.5 |
| DetCLIPv3 [41] | Swin-L | 48.8 | 49.9 | 49.7 | 47.8 | 41.4 | 41.4 | 40.5 | 42.3 |
| LLMDet [11] | Swin-L | 51.1 | 45.1 | 46.1 | 56.6 | 42.0 | 31.6 | 38.8 | 50.2 |
| VL-SAM-V2 (Our) | Swin-L | **51.7** | 47.2 | 46.7 | 56.6 | **42.5** | 33.2 | 39.7 | 50.2 |

Table 2: **Comparison of zero-shot open-ended object detection results on LVIS minival [13].** We report *fixed* AP [8]. VL-SAM utilizes ViT-H [10] of SAM as the image encoder for segmentation.

| Method | Image Encoder | Vision-Language Model | AP | AP$_{rare}$ |
|---|---|---|---|---|
| GenerateU [25] | Swin-T | FlanT5-base | 26.8 | 20.0 |
| VL-SAM-V2 (Our) | Swin-T | InternVL-2.5 (8B) | 29.5 | 29.8 |
| GenerateU [25] | Swin-L | FlanT5-base | 27.9 | 22.3 |
| VL-SAM [27] | ViT-H | CogVLM (17B) | 25.3 | 23.4 |
| VL-SAM-V2 (Our) | Swin-L | InternVL-2.5 (8B) | 31.8 | 30.5 |

on frequent objects. In contrast, VL-SAM-V2 obtains both the best AP and AP$_r$, demonstrating that the fusion with the general proposals from open-set models and specific proposals from vision-language models helps the models to retain the high performance of open-set models on frequent objects.

### 4.3 Ablation Study

In this section, we conduct ablation experiments to analyze the main components and model generalization of VL-SAM-V2. We use Swin-T as the backbone and report the results on LVIS minival.

**Main Components.** To evaluate the effectiveness of each component, we successively add components to the LLMDet baseline. As shown in Table 4, we observe that each component consistently improves performance. Specifically, adding general and specific query fusion brings improvements of 0.5 AP, 1.8 AP$_r$, and 0.8 AP$_c$, respectively. Then, we adopt ranked learnable queries to replace the single learnable query for each point prompt, obtaining a 1.1 AP$_r$ improvement. Moreover, training with denoising points can further improve AP$_r$ by 1.0. Meanwhile, the overall training cost is reduced by 50% due to the removal of point generation in VL-SAM.

Table 3: **Comparison of open-set object detection results on COCO and CODA.** We report mAP for each dataset.

| Method | COCO | CODA |
|---|---|---|
| GroundingDINO | 48.4 | 12.6 |
| YOLO-World | 45.1 | 16.1 |
| LLMDet | 55.6 | 38.5 |
| VL-SAM-V2 | 56.0 | 42.3 |

Table 4: **Ablations on main components.** 'GS Fusion' denotes the general and specific query fusion module. Each component improves the detection performance consistently.

| GS Fusion | Ranked Learnable Queries | Denoising Points | AP | $AP_r$ | $AP_c$ |
|---|---|---|---|---|---|
| | | | 44.7 | 37.3 | 39.5 |
| ✓ | | | 45.2 | 39.1 | 40.3 |
| ✓ | ✓ | | 45.4 | 40.2 | 40.6 |
| ✓ | ✓ | ✓ | 45.7 | 41.2 | 41.1 |

Table 5: **Ablations on fusion methods.** 'GS Fusion' denotes the general and specific query fusion module.

| Fusion methods | AP | $AP_r$ |
|---|---|---|
| Naive Concatenate | 44.9 | 38.5 |
| Late Fusion | 45.1 | 40.2 |
| GS Fusion | 45.7 | 41.2 |

Overall, the results demonstrate the effectiveness of each component proposed in VL-SAM-V2.

**Fusion Methods.** In addition to the proposed general and specific query fusion, we try two other fusion methods, *i.e.*, Naive Concatenate and Late Fusion. Naive Concatenate simply concatenates two query features into one feature with a linear layer. Late Fusion directly ensembles the detection results from two queries. As shown in Table 5, we can observe that GS Fusion achieves the best fusion performance, outperforming naive concatenate by 2.7 $AP_r$.

**Model Generalization.** To demonstrate the model generalization of the VL-SAM-V2, we adapt various popular open-set models and vision-language models, including GroundingDINO [29], LLMDet [11], LLaVA [28], CogVLM [38], and InternVL-2.5 [5].

As shown in Table 6, VL-SAM-V2 consistently improves the performance of different open-set models with different vision-language models, ranging from 2.3 to 4.9 $AP_r$. Moreover, the empirical results show that stronger vision-language models can obtain better detection performance, aligning with the finding in VL-SAM [27]. This indicates that VL-SAM-V2 can benefit from more powerful open-set models and vision-language models.

### 4.4 Combined with SAM.

Following Grounded-SAM [35], we combine the proposed VL-SAM-V2 with the segment-anything model (SAM) [18] to achieve open-world instance segmentation.

As shown in Table 7, we compare the proposed method combining SAM with VL-SAM on the open-ended instance segmentation task. We can find that our method shows favorable performance improvement. In addition, we visualize the open-ended instance segmentation results on the corner case object detection dataset, CODA, to demonstrate the effectiveness of the proposed method in real-world scenarios. As depicted in Figure 4, our method can provide dense segmentation masks and category annotations for input images and discover various novel object categories outside of the existing autonomous driving datasets [36, 1], including sacks and cranes. Moreover, since VL-SAM-V2 uses point prompts as inputs, users can provide point prompts by themselves to achieve

Table 6: **Ablation of model generalization.** VL-SAM-V2 can adopt various open-set models and vision-language models for point prompting. † denotes results are reimplemented by MMDetection [2].

| Method | Open-set Models | Vision-Language Model | AP | $AP_{rare}$ |
|---|---|---|---|---|
| GroundingDINO† | - | - | 41.4 | 34.2 |
| LLMDet | - | - | 44.7 | 37.3 |
| VL-SAM-V2 (Our) | GroundingDINO | LLaVA (7B) | 42.1 | 37.3 |
| | | CogVLM (17B) | 42.7 | 38.5 |
| | | InternVL-2.5 (8B) | 43.0 | 39.1 |
| | LLMDet | LLaVA (7B) | 45.2 | 39.6 |
| | | CogVLM (17B) | 45.5 | 40.6 |
| | | InternVL-2.5 (8B) | 45.7 | 41.2 |

Table 7: **Comparison of zero-shot open-ended instance segmentation results on LVIS minival [13].**

| Method | Vision-Language Model | mask AP | mask $AP_{rare}$ |
|---|---|---|---|
| VL-SAM [27] | CogVLM (17B) | 23.9 | 22.7 |
| VL-SAM-V2 (Our) | InternVL-2.5 (8B) | 28.7 | 27.7 |

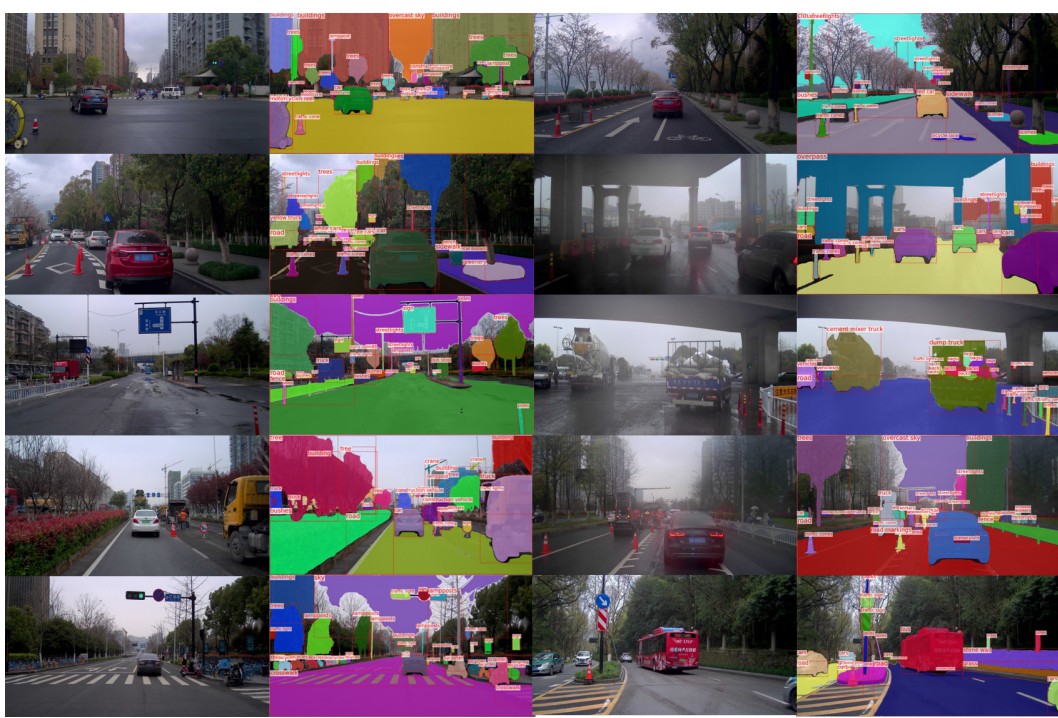

Figure 4: **Visualization results VL-SAM-V2 combining with SAM on CODA [22].** We show input images and detection and segmentation prediction results in the open-ended mode. VL-SAM-V2 can discover various uncommon objects. Best viewed by zooming in.

prompt-based detection and segmentation, demonstrating the diverse applications of the proposed method in real-world scenarios.

# 5 Limitations

Since we adopt the VL-SAM framework to utilize vision-language models to generate point prompts, VL-SAM-V2 inherits the defects of vision-language models, including the hallucination problem, incorrect responses, and low inference speed. However, these defects will be gradually overcome with the development of vision-language models. The proposed VL-SAM-V2 framework can benefit from these new models. Moreover, VL-SAM-V2 needs additional SAM to obtain instance segmentation results. In the future, we will integrate segmentation models into our framework to compose an end-to-end open-world perception model.

# 6 Conclusion

In this work, we introduce VL-SAM-V2, a novel open-world object detection framework that can perform in the open-set or open-ended mode. The core of VL-SAM-V2 is the general and specific query fusion that integrates information from both open-set and open-ended perception paradigms. To further enhance the model's ability to discover unseen objects, we present ranked learnable queries to promote more diverse and representative query representations. Additionally, we propose the denoising point training strategy to stabilize and facilitate the training process. Experimental results on the LVIS demonstrate that VL-SAM-V2 outperforms previous state-of-the-art methods in both open-set and open-ended settings, particularly excelling in detecting rare objects. Moreover, VL-SAM-V2 exhibits good model generalization that can incorporate various open-set models and vision-language models. By combining with SAM, VL-SAM-V2 shows promise for a new generation of automatic labeling and open-world perception systems.

**Broader Impacts Statement.** This paper investigates the use of current open-set models and vision-language models for open-world object detection. We do not see potential privacy-related issues. This study may inspire future research on automatic labeling systems and open-world perception models.

**Acknowledgments.** This work was supported by National Key R&D Program of China (Grant No. 2022ZD0160305).

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
