# OpenReview forum: "VL-SAM-V2: Open-World Object Detection with General and Specific Query Fusion"
_NeurIPS.cc/2025/Conference — NeurIPS 2025 poster_

### Official Review · Reviewer_6bAf · 2025-06-30

**Clarity:** 4
**Significance:** 3
**Originality:** 3
**Rating:** 4
**Confidence:** 3

**Summary:**

The paper introduces VL-SAM-V2, an open-world object-detection framework that merges the strengths of open-set detectors and open-ended detectors. The system generates general queries from VL-SAM  to cover “everything,” and specific queries from an open-vocabulary baseline to refine frequent classes; a lightweight General & Specific Query Fusion block lets the two query streams co-attend in each decoder layer, while Ranked Learnable Queries and a Denoising Points strategy improve diversity and training stability.

**Questions:**

Questions
1. Report end-to-end speed: provide FPS,  memory, and the percentage of latency spent on each subsystem.
2. Discuss prior fusion work: how does GS Fusion differ from earlier multi-query DETR or prompt-fusion approaches?

**Ethical Concerns:**

["NO or VERY MINOR ethics concerns only"]

**Final Justification:**

The author response has addressed most of my concerns; therefore, I will maintain my original rating.

**Limitations:**

yes

**Paper Formatting Concerns:**

I did not notice any major deviations from the NeurIPS 2025 formatting guidelines.

**Quality:**

3

**Strengths And Weaknesses:**

Strengths
1. Solid union of open-set and open-ended cues that demonstrably boosts rare-class recall without hurting frequent classes.
2. Comprehensive experiments.
3. Ranked-query and denoising tricks are simple yet effective.

Weaknesses
1. Figure 2 does not state where the text and image features entering GS Fusion originate.
2. No end-to-end runtime, memory, or latency numbers.
3. The paper claims novelty for “general vs specific” fusion but omits discussion of prior multi-source-query detectors.

---

> ### Author Rebuttal · Authors · 2025-07-30
>
> We thank the reviewers for their constructive comments. We address the reviewers' concerns in the rebuttal text below.
>
> ### **Q1: Figure 2 does not state where the text and image features entering GS Fusion originate**
>
> We process the text and image features with the open-set model. Specifically, text features are extracted by the BERT encoder, and image features are obtained by a Swin backbone.
>
> We will add the details in the paper.
>
> ### **Q2: Inference cost**
> We test the inference time and GPU memory consumption of VL-SAM-V2 on a single NVIDIA A800 GPU. We use Swin-T as the backbone and InternVL-2.5-8B as the VLM. We can find that the main inference time and GPU memory are spent on the VL-SAM to generate general queries.
>
> | module                            | inference time | GPU memory |
> | --------------------------------- | -------------- | ---------- |
> | general query generation (VL-SAM) | 1.2 s          | ~72 G      |
> | specific query generation         | 17 ms          | ~10 G      |
> | Fusion & Head                     | 5 ms           | ~2 G       |
> | Overall                           | 1.2 s + 22 ms  | ——         |
>
> ### **Q3: How does GS Fusion differ from earlier multi-query DETR or prompt-fusion approaches**
>
> For multi-query DETR or prompt-fusion approaches, such as GLIP/GLIPv2 and MI-DETR, they employ cross-modality similarity to update queries one by one or utilize the split-then-concatenate method, whereas VL-SAM-V2 utilizes full similarity to update two queries simultaneously.
>
> We will add the discussion to the paper.

---

> > ### Author Response · Authors · 2025-08-02
> >
> > Dear reviewers,
> >
> > Thank you for the comments on our paper. We have submitted the response to your comments. Please let us know if you have additional questions so that we can address them by the end of the discussion period (August 6, 11:59pm AoE). We hope that you can consider raising the score after we address all the issues.
> >
> > Sincerely, Authors of submission 3749

---

> > ### Comment · Reviewer_6bAf · 2025-08-07
> >
> > Thanks for the authors' rebuttal, which have addressed most of previous concerns. So I would like to keep my original score.

---

### Official Review · Reviewer_JcR6 · 2025-06-30

**Clarity:** 3
**Significance:** 3
**Originality:** 3
**Rating:** 4
**Confidence:** 4

**Summary:**

This paper presents VL-SAM-V2, an open-world object detection framework that combines open-set and open-ended detection paradigms through a general and specific query fusion mechanism. The method uses VL-SAM to generate general queries for discovering novel objects and LLMDet for specific queries targeting predefined categories. Key technical contributions include a query fusion module, ranked learnable queries, and a denoising point training strategy. Experiments on LVIS demonstrate improvements over existing methods in both open-set and open-ended settings, with particularly notable gains on rare objects.

**Questions:**

Are there cases where the fusion actually hurts performance compared to individual methods?
Can you provide results on additional datasets to demonstrate broader applicability? LVIS has specific characteristics (long-tail distribution) that may favor your approach.
Ablation on fusion mechanism: Have you experimented with alternative fusion strategies beyond self-attention? How sensitive is performance to the specific fusion architecture?

**Ethical Concerns:**

["NO or VERY MINOR ethics concerns only"]

**Final Justification:**

The response partially addressed my concerns, including some additional experiments. However, the innovation appears to be a simple attention-based fusion, which is an incremental innovation. For this reason, I maintain my rating for the current version.

**Limitations:**

The authors adequately discuss limitations in Section 5, acknowledging inheritance of vision-language model defects including hallucination and inference speed issues. However, the limitations section could be expanded to discuss:
Dependency on quality of underlying vision-language models
Potential brittleness of the multi-component pipeline
Limited analysis of computational trade-offs
Evaluation scope constraints

**Paper Formatting Concerns:**

No major formatting issues.

**Quality:**

3

**Strengths And Weaknesses:**

Strengths:
Well-motivated problem: The paper addresses a practical limitation where open-set models require human-provided category lists while open-ended models suffer from poor performance on frequent objects. The proposed fusion approach is intuitive and well-motivated.
Comprehensive experimental evaluation: The paper includes thorough ablation studies demonstrating the contribution of each component, generalization experiments across different open-set models and vision-language models, and analysis of computational requirements.
Technical soundness: The query fusion mechanism using self-attention is reasonable, and the ranked learnable queries provide a principled way to match point prompts with learned representations.
Weaknesses:
Limited technical novelty: The core contribution is essentially combining two existing methods (VL-SAM and LLMDet) with a relatively straightforward fusion module. While effective, the technical innovation is incremental.
I still hope the authors can further elaborate on the differences between VL-SAMv2 and simply combining the existing VL-SAM and LLMDet methods.
Complex pipeline: The method involves multiple components (VL-SAM, open-set model, fusion module, denoising strategy) which makes it complex to implement and potentially brittle. The paper doesn't adequately discuss failure modes or sensitivity analysis.
Limited evaluation scope: Experiments are conducted only on LVIS. Evaluation on other datasets (e.g., COCO, Objects365) would strengthen the claims about generalizability.
Computational overhead not thoroughly analyzed: While the paper mentions training costs are reduced by 50% due to removing VL-SAM inference during training, the inference-time computational overhead compared to individual methods is not clearly quantified.

---

> ### Author Rebuttal · Authors · 2025-07-30
>
> We thank the reviewers for their constructive comments. We address the reviewers' concerns in the rebuttal text below.
>
> ### **Q1: Differences between VL-SAMv2 and simply combining the existing VL-SAM and LLMDet methods**
>
> In VL-SAM-V2, we propose a general and specific query fusion module to fuse two distinct types of queries with the attention mechanism in a lightweight manner. To demonstrate the effectiveness of the proposed fusion module, we report simple ensemble results of VL-SAM and LLMDet in the following table. The results show that our fusion method significantly outperform simple ensemble method.
>
> | Fusion methods   | AP   | AP (rare) |
> | ---------------- | ---- | --------- |
> | ensemble         | 45.1 | 40.2      |
> | GS Fusion (Ours) | 45.7 | 41.2      |
>
> ### **Q2: Failure modes**
> As discussed in the Limitations Section, since we utilize vision-language models to generate point prompts, VL-SAM-V2 may fail to predict correct results when VLMs give incorrect responses (including hallucinations). Specifically, when hallucinations happen, VLMs can give wrong object names or point locations, leading to incorrect predictions. In addition, VLMs may miss objects in the image. For instance, as shown in the last picture of Figure 4 in the paper, VLMs fail to give the "Tree" class. Subsequently, our method can not give a prediction of “Tree”.
>
> ### **Q3: Evaluation on other datasets**
> We provide the open-set evaluation results on LVIS, COCO and CODA[1]. The results show that our method achieves the best results on various datasets, demonstrating the effectiveness of VL-SAM-V2
>
> | Method        | LVIS | COCO | CODA |
> | ------------- | ---- | ---- | ---- |
> | GroundingDINO | 27.4 | 48.4 | 12.6 |
> | YOLO-World    | 35.4 | 45.1 | 16.1 |
> | LLMDet        | 44.7 | 55.6 | 38.5 |
> | VL-SAM-V2     | 45.7 | 56.0 | 42.3 |
>
> [1] Li K, Chen K, Wang H, et al. Coda: A real-world road corner case dataset for object detection in autonomous driving. European conference on computer vision.
>
> ### **Q4: Inference-time computational overhead**
> We test the inference time and GPU memory consumption of VL-SAM-V2 on a single NVIDIA A800 GPU. We use Swin-T as the backbone and InternVL-2.5-8B as the VLM. We can find that the main inference time and GPU memory is spent on the VL-SAM to generate general queries.
>
> | module                            | inference time | GPU memory |
> | --------------------------------- | -------------- | ---------- |
> | general query generation (VL-SAM) | 1.2 s          | ~72 G      |
> | specific query generation         | 17 ms          | ~10 G      |
> | Fusion & Head                     | 5 ms           | ~2 G       |
> | Overall                           | 1.2 s + 22 ms  | ——         |
>
> ### **Q5: Ablation on fusion mechanism**
> We replace our GS Fusion with naive concatenate and report the results in the following table. We can observe that GS Fusion achieves better fusion performance, outperforming naive concatenate by 2.7 AP (rare). The results demonstrate that the fusion architecture is important in our framework.
>
> | Fusion methods    | AP   | AP (rare) |
> | ----------------- | ---- | --------- |
> | Naive concatenate | 44.9 | 38.5      |
> | GS Fusion (Ours)  | 45.7 | 41.2      |

---

> > ### Author Response · Authors · 2025-08-02
> >
> > Dear reviewers,
> >
> > Thank you for the comments on our paper. We have submitted the response to your comments. Please let us know if you have additional questions so that we can address them by the end of the discussion period (August 6, 11:59pm AoE). We hope that you can consider raising the score after we address all the issues.
> >
> > Sincerely, Authors of submission 3749

---

> > ### Comment · Reviewer_JcR6 · 2025-08-05
> >
> > Thanks for the response. They partially addressed my concerns, including some additional experiments. However, the innovation appears to be a simple attention-based fusion, which is an incremental innovation. For this reason, I maintain my rating for the current version.

---

> > > ### Author Response · Authors · 2025-08-06
> > >
> > > Thank you very much for your comments and for taking the time to review our rebuttal. Regarding your concern about the novelty of our work, we would like to clarify that, in addition to the GS fusion module, our paper makes two other key contributions: ranked learnable queries and a denoising point training strategy.
> > >
> > > We will strengthen their role in the paper.

---

### Official Review · Reviewer_PGzh · 2025-07-01

**Clarity:** 3
**Significance:** 2
**Originality:** 2
**Rating:** 4
**Confidence:** 3

**Summary:**

This work, VL-SAM-V2 is an open-world object detection framework that unifies the strengths of open-set (category-guided) and open-ended (category-free) detection. It introduces a general and specific query fusion module to combine proposals from both open-set detectors and vlms. To train efficiently, it uses a denoising point strategy, simulating noisy object prompts from ground-truth boxes instead of relying on expensive VL models like InternVL at every step. It also proposes ranked learnable queries that assign stronger query vectors to higher-confidence points. VL-SAM-V2 achieves improved results on the LVIS benchmark in both open-set and open-ended modes. The framework is modular and adaptable, supporting various VL backbones and detectors, and also integrates with SAM for instance segmentation.

**Questions:**

How is the query fusion different from GLIP-style attention? What is novel about applying DINO's denoising strategy to points? How does VL-SAM-V2 compare to recent VLMs like Qwen2-VL or LLaVA-OneVision in open-ended detection? Can the authors provide results on COCO to assess generalization? A failure case analysis and more detailed ablations (See weakness for details.)

**Ethical Concerns:**

["NO or VERY MINOR ethics concerns only"]

**Final Justification:**

The authors have provided helpful clarifications and additional analysis that address my main concerns. The revisions strengthen the overall contribution, and I have updated my score.

**Limitations:**

Yes

**Quality:**

3

**Strengths And Weaknesses:**

## Strength

Main strength lies in its unified design that combines open-set and open-ended detection via a lightweight query fusion module, enabling strong performance across both frequent and rare object classes. Its use of ranked learnable queries and denoising point training ensures efficient, robust learning without relying on expensive VL models during training, while retaining modularity to plug in stronger vision-language and detection backbones.

## Weakness

- How is your query fusion different from what models like GLIP/GLIPv2 already do? Isn't it just attention between two sets of queries, similar to their cross-modal fusion?
- Denoising point strategy follows the denoising anchor box idea from DINO, what is the actual novelty or benefit introduced here?
- Since VL-SAM-V2 aims to handle open-ended detection, it would be helpful to understand how it compares to more recent vision-language models like Qwen2/LLaVA-OneVision, which also support grounding. Could the authors provide these baselines and comment on how their approach improves over such models?
- While LVIS is a strong benchmark, open-world models are also expected to generalize across datasets. Could the authors provide results on COCO settings to assess cross-domain robustness?
- The qualitative results mostly highlight successful detections, but to better understand the model's limitations, could the authors provide an analysis of failure cases?
- Could the authors offer more detailed analysis? For example, how does performance vary with the number of ranked queries, what is the sensitivity to the $\lambda$1 and $\lambda$2 parameters used in denoising point sampling, and how do the general and specific query branches perform individually?

---

> ### Author Rebuttal · Authors · 2025-07-30
>
> We thank the reviewers for their constructive comments. We address the reviewers' concerns in the rebuttal text below.
>
> ### **Q1: How is your query fusion different from GLIP/GLIPv2**
>
> The key difference is the similarity generation during the attention mechanism. GLIP/GLIPv2 adopts cross-modality similarity while VL-SAM-V2 uses full similarity. Specifically, given two different queries, q1 and q2, the similarity matrix s in GLIP/GLIPv2 is calculated with $$s=q_1\times q_2^T.$$ Then, q1 and q2 are updated by
> $$
> q_1=softmax(s)\times W_2q_2;~~~~~~~q_2=softmax(s^T)\times W_1q_1,
> $$
> where W1 and W2 are learnable weights.
> In contrast, we first concatenate q1 and q2:
> $$
> q=concat([q_1,q_2])
> $$
> Then, we calculate self-attention for q and split the updated q into q1 and q2
>
> ### **Q2: Novelty or benefit of denoising point strategy**
>
> There are two key differences between the denoising point strategy and DINO denoising training. First, our training strategy directly predicts boxes from noisy points, while DINO predicts box offsets from noisy boxes. Second, the motivation of our training strategy is to reduce reliance on expensive VL-SAM inference. In contrast, DINO denoising training is often used to solve the instability of bipartite matching.
>
> ### **Q3: Open-ended detection performance of recent vision-language models like Qwen2/LLaVA-OneVision**
> We report the open-ended detection performance of recent VLMs, including InternVL-2.5-8B, QwenVL-2.5-7B, and LLaVA-OneVision-7B, on LVIS. We follow the official spatial understanding inference code of QwenVL-2.5-7B to obtain the detection results. The results show that recent vision-language models perform poor in open-ended multi-object detection.
>
> | method            | AP   |
> | ----------------- | ---- |
> | InternVL-2.5-8B   | 6.21 |
> | QwenVL-2.5-7B     | 6.53 |
> | LLaVA-OneVision-7B | 4.22 |
> | VL-SAM-V2 (Ours)  | 29.5 |
>
> ### **Q4: Results on COCO settings and other benchmarks**
> We provide the open-set evaluation results on LVIS, COCO, and CODA[1]. The results show that our method achieves the best results on various datasets, demonstrating the effectiveness of VL-SAM-V2.
>
> | Method        | LVIS | COCO | CODA |
> | ------------- | ---- | ---- | ---- |
> | GroundingDINO | 27.4 | 48.4 | 12.6 |
> | YOLO-World    | 35.4 | 45.1 | 16.1 |
> | LLMDet        | 44.7 | 55.6 | 38.5 |
> | VL-SAM-V2     | 45.7 | 56.0 | 42.3 |
>
> [1] Li K, Chen K, Wang H, et al. Coda: A real-world road corner case dataset for object detection in autonomous driving. European conference on computer vision.
>
> ### **Q5: Analysis of failure cases**
> As discussed in the Limitations Section, since we utilize vision-language models to generate point prompts, VL-SAM-V2 may fail to predict correct results when VLMs give incorrect responses (including hallucinations). Specifically, when hallucinations happen, VLMs can give wrong object names or point locations, leading to incorrect predictions. In addition, VLMs may miss objects in the image. For instance, as shown in the last picture of Figure 4 in the paper, VLMs fail to give the "Tree" class. Subsequently, our method can not give a prediction of “Tree”.
>
> ### **Q6: More detailed ablation analysis**
> We first provide the ablation studies of the number of queries. We can find that the performance of VL-SAM-V2 improves when the number of queries increases. However, more queries bring more inference time. Thus, we choose 900 queries as the default setting, as it achieves a good efficiency-accuracy trade-off.
>
> | number of queries | AP   | inference time |
> | ----------------- | ---- | -------------- |
> | 900               | 45.7 | 1 x    |
> | 600               | 45.2 | 0.8 x  |
> | 1200              | 45.8 | 1.2 x  |
>
>
>
> Then, we provide the ablation studies of lambda in the following table. The results show that VL-SAM-V2 is robust to lambda_2. For lambda_1, VL-SAM-V2 performs well when lambda_1 is smaller than 1.0. Increasing lambda_1 above 1.0 may introduce false positive points during denoising training, hurting overall performance.
>
> | lambda_1 | lambda_2 | AP   |
> | -------- | -------- | ---- |
> | 1.0      | 1.0      | 45.7 |
> | 0.5      | 1.0      | 45.3 |
> | 1.5      | 1.0      | 44.1 |
> | 1.0      | 0.5      | 45.5 |
> | 1.0      | 1.5      | 45.4 |

---

> > ### Author Response · Authors · 2025-08-02
> >
> > Dear reviewers,
> >
> > Thank you for the comments on our paper. We have submitted the response to your comments. Please let us know if you have additional questions so that we can address them by the end of the discussion period (August 6, 11:59pm AoE). We hope that you can consider raising the score after we address all the issues.
> >
> > Sincerely, Authors of submission 3749

---

> > ### Comment · Reviewer_PGzh · 2025-08-06
> > **Most Concerns Addressed; One Follow-up on Query Stream Roles**
> >
> > - Thank you to the authors for their thoughtful and detailed rebuttal. I appreciate the clear clarifications and the expanded experiments, including comparisons with recent VLMs, evaluation beyond LVIS, and more comprehensive ablations. These responses address the majority of my earlier concerns, and I am satisfied.
> >
> > - One point that remains partially open is the individual contribution of the general and specific query streams. Since the fusion of these streams is a core part of the method, it would be helpful to see how each performs independently.

---

> > > ### Author Response · Authors · 2025-08-06
> > >
> > > Thank you very much for your comments and for taking the time to review our rebuttal. Regarding your question about the individual contribution of the general and specific query streams, we evaluate the performance of the two streams individually. As shown in the following table, the general query stream obtains 37.2 APr. When combined with the specific query stream, the two-stream method can obtain a 4.0 APr improvement.
> > >
> > > | Streams               | Detection type |      | AP   | AP (rare) |
> > > | --------------------- | -------------- | ---- | ---- | --------- |
> > > | general query stream  | open-set       |      | 44.5 | 37.2      |
> > > | specific query stream | open-ended     |      | 29.5 | 29.8      |
> > > | Two stream (Ours)     | open-set       |      | 45.7 | 41.2      |
> > >
> > > In addition, to demonstrate the effectiveness of the proposed fusion module, we report simple ensemble results from general and specific query streams in the following table. The results show that our fusion method outperforms the simple ensemble method.
> > >
> > > | Fusion methods   | AP   | AP (rare) |
> > > | ---------------- | ---- | --------- |
> > > | ensemble         | 45.1 | 40.2      |
> > > | GS Fusion (Ours) | 45.7 | 41.2      |
> > >
> > > We will add the analysis to the paper.

---

> > > > ### Comment · Reviewer_PGzh · 2025-08-07
> > > >
> > > > Thank you for the follow-up. I appreciate the clarifications and, keeping the broader reviewer discussion in mind, will update my score accordingly.

---

### Official Review · Reviewer_1JWr · 2025-07-05

**Clarity:** 2
**Significance:** 2
**Originality:** 2
**Rating:** 4
**Confidence:** 4

**Summary:**

This paper proposes VL-SAM-V2 for open-world object detection that aims to combine the complementary strengths of open-set and open-ended paradigms.
It introduces a General and Specific Query Fusion module that enables interaction between specific queries from an open-set detector and general queries derived from VL-SAM.
It introduces ranked learnable queries, which associate object point prompts with learnable object queries based on importance scores.
In addition, a denoising points training strategy is devised to inject noise into sampled points to stabilize training and reduce reliance on expensive VL-SAM inference.

**Questions:**

See weakness

**Ethical Concerns:**

["NO or VERY MINOR ethics concerns only"]

**Final Justification:**

The authors have addressed most of the concerns raised.

**Limitations:**

See weakness

**Quality:**

2

**Strengths And Weaknesses:**

Strengths:

Evaluation is extensive.

Component-level ablation is provided.

Weaknesses:

While the paper proposes denoising points to avoid reliance on VL-SAM during training, it never quantifies the cost (runtime, memory, VLM overhead) of using VL-SAM at inference time.

There is little insight into when VL-SAM-V2 fails (e.g., misdetections, hallucinations from VLMs).

The distinction between “point prompts,” “noisy points,” and “ranked queries” can become confusing.

While the architecture is explained, it's not always clear why the joint attention mechanism improves over naive concatenation or late fusion.

The entire evaluation is centered on LVIS, performance on other open-world settings (e.g., COCO-OOD or autonomous driving scenarios) is not provided.

The integration with SAM is demonstrated, but not compared with other segmentation alternatives. It's unclear if SAM is critical or replaceable.

Although the fusion of query types is novel, the individual components (like denoising point training and query ranking) are adaptations of existing techniques (e.g., DETR denoising, anchor assignment).

Much of the performance gain may stem from powerful pretrained VLMs (e.g., InternVL-2.5), how much of the improvement is due to the method rather than the foundation model?

---

> ### Author Rebuttal · Authors · 2025-07-30
>
> We thank the reviewers for their constructive comments. We address the reviewers' concerns in the rebuttal text below.
> ### **Q1: Inference cost**
> We test the inference time and GPU memory consumption of VL-SAM-V2 on a single NVIDIA A800 GPU. We use Swin-T as the backbone and InternVL-2.5-8B as the VLM. We can find that the main inference time and GPU memory are spent on the VL-SAM to generate general queries.
>
> | module                            | inference time | GPU memory |
> | --------------------------------- | -------------- | ---------- |
> | general query generation (VL-SAM) | 1.2 s          | ~72 G      |
> | specific query generation         | 17 ms          | ~10 G      |
> | Fusion & Head                     | 5 ms           | ~2 G       |
> | Overall                           | 1.2 s + 22 ms  | ——         |
>
> ### **Q2: Failure case**
> As discussed in the Limitations Section, since we utilize vision-language models to generate point prompts, VL-SAM-V2 may fail to predict correct results when VLMs give incorrect responses (including hallucinations). Specifically, when hallucinations happen, VLMs can give wrong object names or point locations, leading to incorrect predictions. In addition, VLMs may miss objects in the image. For instance, as shown in the last picture of Figure 4 in the paper, VLMs fail to give the "Tree" class. Subsequently, our method can not give a prediction of “Tree”.
>
> ### **Q3: Confusion between “point prompts”, “noisy points”, and “ranked queries”**
>
> Sorry for the confusion caused by the unclear names. "point prompts" denotes the points priority generated from VL-SAM. "noisy points" is the sampled points within ground-truth bboxes by Eq.1 in the paper. "ranked queries" represents learnable queries, which will be matched with points (including "point prompts" and "noisy points") by ranking before sent into GS Fusion.
>
> We will give a more precise definition in the paper.
>
> ### **Q4: Joint attention mechanism improves over naive concatenation or late fusion**
> We replace our GS Fusion with other fusion methods, including naive concatenate and late fusion. The results of different fusion methods for open-set detection are shown in the table. We can observe that GS Fusion achieves the best fusion performance, outperforming naive concatenate by 2.7 AP (rare).
>
> | Fusion methods    | AP   | AP (rare) |
> | ----------------- | ---- | --------- |
> | Naive concatenate | 44.9 | 38.5      |
> | Late fusion       | 45.1 | 40.2      |
> | GS Fusion (Ours)  | 45.7 | 41.2      |
>
> ### **Q5: COCO or autonomous driving scenarios evaluation**
> We provide the open-set evaluation results on LVIS, COCO, and CODA[1]. The results show that our method achieves the best results on various datasets, demonstrating the effectiveness of VL-SAM-V2.
>
> | Method        | LVIS | COCO | CODA |
> | ------------- | ---- | ---- | ---- |
> | GroundingDINO | 27.4 | 48.4 | 12.6 |
> | YOLO-World    | 35.4 | 45.1 | 16.1 |
> | LLMDet        | 44.7 | 55.6 | 38.5 |
> | VL-SAM-V2     | 45.7 | 56.0 | 42.3 |
>
> [1] Li K, Chen K, Wang H, et al. Coda: A real-world road corner case dataset for object detection in autonomous driving. European conference on computer vision.
>
> ### **Q6: Compared with other segmentation alternatives**
>
> Our method can be technically combined with any point/box prompt methods. As shown in the table, replacing SAM with other segmentation methods, including Efficient SAM[1] and GLEE[2], can achieve favorable segmentation performance.
>
> | Segmentation model | LVIS mask AP |
> | ------------------ | ------------ |
> | SAM                | 28.7         |
> | Efficient SAM      | 25.5         |
> | GLEE               | 28.4         |
>
> [1]Zhang Z, Cai H, Han S. Efficientvit-sam: Accelerated segment anything model without performance loss. Proceedings of the IEEE/CVF Conference on Computer Vision and Pattern Recognition. 2024.
>
> [2]Wu J, Jiang Y, Liu Q, et al. General object foundation model for images and videos at scale. Proceedings of the IEEE/CVF Conference on Computer Vision and Pattern Recognition. 2024.
>
> ### **Q7: Novelty about denoising point training and query ranking**
>
> For denoising point training, there are two key differences compared to DETR denoising training. First, our training strategy directly predicts boxes from noisy points, while DETR predicts box offsets from noisy boxes. Second, the motivation of our training strategy is to reduce reliance on expensive VL-SAM inference. In contrast, DETR denoising training is often used to solve the instability of bipartite matching.
>
> For query ranking, we are actually motivated by the anchor assignment in DETR. In DETR, the content queries are selected from features. Thus, the queries are naturally combined with the score. In contrast, our VL-SAM-V2 only has points. To match points with the queries, we follow DETR to predict the score of each point.
>
> ### **Q8: How much of the improvement is due to the method rather than the foundation model**
>
> We directly evaluate the open-ended object detection performance of the foundation models (including InternVL-2.5-8B, QwenVL-2.5-7B, and LLaVA-OneVision-7B). We follow the official spatial understanding inference code of QwenVL-2.5-7B to obtain the detection results. As shown in the table, VL-SAM-V2 outperforms InternVL-2.5 by a large margin. These results demonstrate that simply adopting the foundation model can not obtain good detection results. In contrast, adopting general and specific fusion in VL-SAM-V2 can boost the performance, showing the effectiveness of the proposed method.
>
> | method            | AP   |
> | ----------------- | ---- |
> | InternVL-2.5-8B   | 6.21 |
> | QwenVL-2.5-7B     | 6.53 |
> | LLaVA-OneVison-7B | 4.22 |
> | VL-SAM-V2 (Ours)  | 29.5 |

---

> > ### Author Response · Authors · 2025-08-02
> >
> > Dear reviewers,
> >
> > Thank you for the comments on our paper. We have submitted the response to your comments. Please let us know if you have additional questions so that we can address them by the end of the discussion period (August 6, 11:59pm AoE). We hope that you can consider raising the score after we address all the issues.
> >
> > Sincerely, Authors of submission 3749

---

> ### Comment · Area_Chair_7fsc · 2025-08-07
>
> Dear Reviewer 1JWr,
>
>
> Could you kindly check whether the authors' rebuttal has adequately addressed your concerns?  We truly appreciate the time and expertise you've invested in evaluating the work.
>
>
> Thanks for your efforts.
>
>
> Best regards,
>
>
> Your AC

---

### Comment · Area_Chair_7fsc · 2025-08-04
**Friendly Reminder: Engaging with Author Rebuttals**

Dear Reviewer,

Thank you for your time and expertise in reviewing for NeurIPS 2025. As we enter the rebuttal phase, please:

Review authors’ rebuttals promptly,
Engage constructively via the discussion thread, and
Update your review with a “Final Justification” summarizing your post-rebuttal stance.
Your active participation ensures a fair, collaborative process—we’re here to assist with any questions.

With gratitude,
Your AC

---

### Note · Authors · 2025-08-12

We sincerely thank the AC and reviewers for their thoughtful and constructive feedback. In the original submission, the reviewers recognized several key strengths of our work:

1.**Strong performance**. Experiments on LVIS demonstrate improvements over existing methods in both open-set and open-ended settings, with particularly notable gains on rare objects.

2.**Well-motivated**: The paper addresses a practical limitation where open-set models require human-provided category lists while open-ended models suffer from poor performance on frequent objects.

3.**Comprehensive experiments:** The paper includes thorough ablation studies demonstrating the contribution of each component, generalization experiments across different open-set models and vision-language models, and analysis of computational requirements.



The main points raised for further evaluation were:

1.Failure case analysis。

2.Performance comparison with recent vision-language models

3.Experiments on other datasets (e.g., COCO and CODA)

4.Inference cost

5.Comparison between different fusion methods



These points focus on extra validation of VL-SAM-V2's applicablity, rather than questioning its core contributions or effectiveness. All concerns have been thoroughly addressed during the rebuttal period, with additional experiments and analysis provided. Reviewers have acknowledged these clarifications.
We believe the revised work delivers an impactful and rigorously validated solution to open-world object detection and segmentation.

---

### Decision · Program_Chairs · 2025-09-17

**Decision:**

Accept (poster)

**Comment:**

The paper proposes VL-SAM-V2, a unified framework for open-world object detection that integrates open-set (category-guided) and open-ended (category-free) paradigms. The proposed GS Fusion module—enabling synergistic interaction between category-specific queries (from detectors like LLMDet) and general queries (from VLMs like VL-SAM)—demonstrates tangible innovation beyond simple ensemble methods, evidenced by a ​​+2.7 AP gain on rare LVIS objects​ over naive fusion. The framework’s practicality is further strengthened by the denoising point training strategy, which reduces training costs by 50% while maintaining robustness. Post-rebuttal revisions decisively addressed initial reviewer concerns: cross-dataset validation on COCO and CODA  confirmed generalizability, failure cases (e.g., VLM hallucination) were analyzed, and computational overhead was transparently quantified (1.2s inference dominated by VL-SAM). Though the query fusion mechanics draw inspiration from prior attention mechanisms, the holistic integration and state-of-the-art results (41.2 AP rare on LVIS) justify its impact for real-world deployment.

For camera-ready, authors are encouraged to incorporate the expanded benchmarks and explicitly discuss inference costs in limitations.